American Society for Microbiology | mSystems®

# Early Introduction of Plant Polysaccharides Drives the Establishment of Rabbit Gut Bacterial Ecosystems and the Acquisition of Microbial Functions

Charlotte Paës,[a,b] Thierry Gidenne,[a] Karine Bébin,[b] Joël Duperray,[c] Charly Gohier,[d] Emeline Guené-Grand,[e] Gwénaël Rebours,[f] Céline Barilly,[a] Béatrice Gabinaud,[a] Laurent Cauquil,[a] Adrien Castinel,[g] Géraldine Pascal,[a] Vincent Darbot,[a] Patrick Aymard,[a] Anne-Marie Debrusse,[a] ⓘ Martin Beaumont,[a] ⓘ Sylvie Combes[a]

[a]GenPhySE, Université de Toulouse, INRAE, ENVT, Castanet-Tolosan, France
[b]CCPA, Janzé, France
[c]EVIALIS, Lieu dit Talhouët, Saint-Nolff, France
[d]MiXscience, Bruz, France
[e]Wisium, Chierry, France
[f]TECHNA, Couëron, France
[g]GeT-PlaGe, Genotoul, INRAE, Castanet-Tolosan, France

**ABSTRACT** In mammals, the introduction of solid food is pivotal for the establishment of the gut microbiota. However, the effects of the first food consumed on long-term microbiota trajectory and host response are still largely unknown. This study aimed to investigate the influences of (i) the timing of first solid food ingestion and (ii) the consumption of plant polysaccharides on bacterial community dynamics and host physiology using a rabbit model. To modulate the first exposure to solid nutrients, solid food was provided to suckling rabbits from two different time points (3 or 15 days of age). In parallel, food type was modulated with the provision of diets differing in carbohydrate content throughout life: the food either was formulated with a high proportion of rapidly fermentable fibers (RFF) or was starch-enriched. We found that access to solid food as of 3 days of age accelerated the gut microbiota maturation. Our data revealed differential effects according to the digestive segment: precocious solid food ingestion influenced to a greater extent the development of bacterial communities of the *appendix vermiformis*, whereas life course polysaccharides ingestion had marked effects on the cecal microbiota. Greater ingestion of RFF was assumed to promote pectin degradation as revealed by metabolomics analysis. However, transcriptomic and phenotypic host responses remained moderately affected by experimental treatments, suggesting little outcomes of the observed microbiome modulations on healthy subjects. In conclusion, our work highlighted the timing of solid food introduction and plant polysaccharides ingestion as two different tools to modulate microbiota implantation and functionality.

**IMPORTANCE** Our study was designed to gain a better understanding of how different feeding patterns affect the dynamics of gut microbiomes and microbe–host interactions. This research showed that the timing of solid food introduction is a key component of the gut microbiota shaping in early developmental stages, though with lower impact on settled gut microbiota profiles in older individuals. This study also provided in-depth analysis of dietary polysaccharide effects on intestinal microbiota. The type of plant polysaccharides reaching the gut through the lifetime was described as an important modulator of the cecal microbiome and its activity. These findings will contribute to better define the interventions that can be employed for modulating the ecological succession of young mammal gut microbiota.

**KEYWORDS** first food, gut health, gut microbiota, intestinal development, metabolomics, microbiota development, microbiota maturation, polysaccharides, starch, young mammal

Address correspondence to Sylvie Combes, sylvie.combes@inrae.fr.

The authors declare no conflict of interest.

10.1128/msystems.00243-22 **1**

The mammalian gut is colonized by a variety of microorganisms, leading to the concept that the host and its inhabiting microbiota constitute a "superorganism," also called holobiont (1). The symbiotic microbiota is essential for the nutrition and health of the host (2).

The microbial ecosystem evolves concomitantly with the host chronological development (3). Microbiota development is first under the influence of the maternal milk, a substrate containing various microbiota-shaping compounds (4, 5). The introduction of solid food later in life represents a new step in the dynamic construction of the gut microbiota with the ingestion of components resistant to host digestion, such as plant cell walls and specific starches (6). This dietary shift modifies the substrates present in the luminal milieu, leading to dramatic changes in the bacterial population in terms of both diversity and composition (7, 8). As a result of this new gut environment, the microbiota deploys metabolic adaptations with increased capacities to degrade complex plant carbohydrates (3, 9). Later in life, bacterial communities continue their adaptation to the host diet (10). For these reasons, the introduction of solid food and diet composition are considered to be major drivers of the microbial succession in the digestive tract (7, 11).

Recent studies have shown that precocious supply of solid food before weaning can modulate the establishment of intestinal bacterial communities and enzymatic activities in several mammalian species (12–16). As a result, the host–microbiota dialogue at the mucosal interface may be affected, such as in lambs, for which inflammation system (12) and digestive tract development (13) modulations were observed following early solid food supplementation. Although these studies demonstrated the influence of the timing of solid food introduction on the host-microbiota codevelopment, the understanding of how early-life solid food ingestion affects the gut microbiota trajectory and the long-term host response remains insufficient.

Among the substances that influence the gut microbiome of mammals, dietary fibers are known to affect the microbiota in late childhood or adulthood, as they are important substrates for bacterial fermentations (17, 18). The term "dietary fiber" encompasses a variety of complex carbohydrates with different physicochemical properties such as resistant starch, nonstarch polysaccharides from plant walls, or nondigestible oligosaccharides (19). The time required by the developing microbial ecosystems to adapt to specific fibrous carbohydrate substrates remains largely unexplored.

Our approach was to provide a dynamic follow-up of the holobiont, evaluated with various measurements, to provide a comprehensive understanding of the impact of different feeding patterns on mammals. In these species, modeling the effects of early-life exposure to solid food is particularly challenging, as it is difficult to capture juvenile feeding patterns (food and milk) without disrupting mother–offspring interactions. To bypass this issue, we propose herein to use a neonatal rabbit model, characterized by a short contact time with the nursing doe for suckling (less than 5 min every 24h) (20). This experimental design allowed us to track the early ingestion of solid food together with a control of milk ingestion. Moreover, 1-week-old rabbits are capable of consuming solid food simultaneously with large amounts of milk (21), making the newborn rabbit a good model to study the timing of solid food introduction in early life. Finally, rabbits are hindgut fermenters that rely heavily on their gut microbiota for digestion and health, allowing for a relevant study of symbiosis in mammals. As commonly observed in mammals, the rabbit cecal microbiota is dominated by *Bacteroidetes* and *Firmicutes* phyla (22) and followed by others including *Proteobacteria* (23). *Ruminococcaceae* and *Lachnospiraceae* are abundant families of the cecal microbiota, with a distinctive feature being the poor colonization of the gut by *Lactobacilli* (24).

The present study aimed to further investigate the mechanisms by which precocious solid food ingestion affects microbiota establishment, with attention given to the dietary polysaccharides. We investigated how the introduction of solid food affects intestinal maturation by providing solid nutrients to suckling rabbits as of 3 or 15 days of age. We examined the impact of two types of plant polysaccharides (rapidly

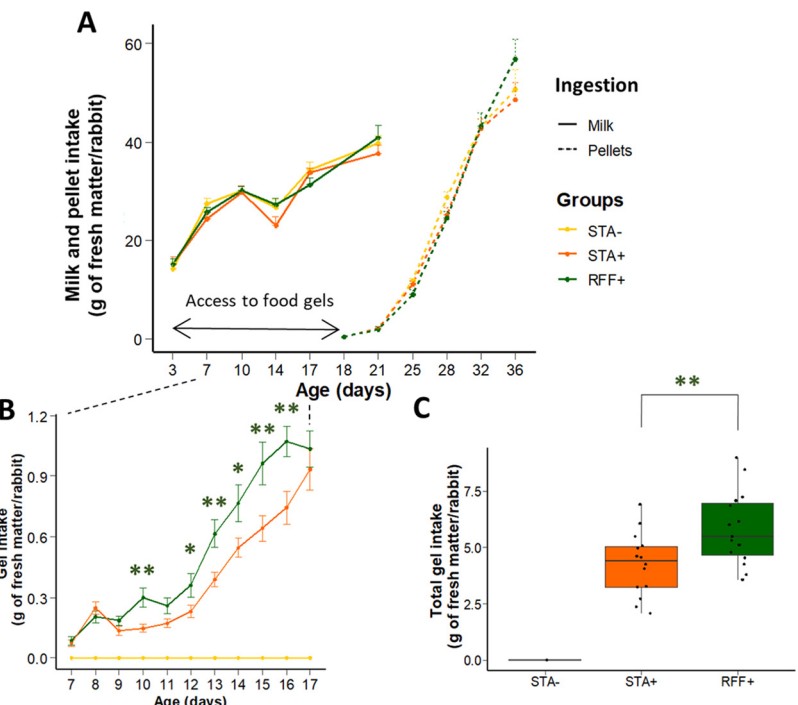

**FIG 1** (A) Suckling rabbits' ingestion pattern (milk and pellets) from d3 to weaning. (B and C) Focus on early consumption of gels in the nest (kinetics and total amounts consumed). Error bars stand for standard error of the mean ($n$ = 15–16 litters of 10 rabbits/group). Green stars emphasize significant differences in gel ingestion between the groups with access to early feeding (STA+/RFF+: *, $P < 0.05$; **, $P < 0.01$). Post-hoc test with correction was used to compare the mean values of each group at different dates.

fermentable fibers [RFF] and starch) consumed during suckling and thereafter, on the gut microbiota. The effects of early solid feeding and plant polysaccharides ingestion on bacterial communities were assessed at five developmental stages in two intestine sections with distinct physiological functions: the cecum, which contributes greatly to host nutrition (25), and the *appendix vermiformis*, a specialized lymphoid organ at the apex of the cecum (26). Extensive investigation of the cecal ecosystem was then performed with the assessment of predictive functional profiling and quantitative metabolic signatures. Gene expression in this tissue was analyzed as a proxy of gut health.

Our findings highlighted that the bacterial communities of rabbit pups quickly responded to the precocious ingestion of solids with both taxonomic and metabolic changes. Although starter feeding resulted in an acceleration of the gut microbiota toward a steady state and increased acetate and butyrate levels early in life, impacts on microbiota activities and host remained moderate after weaning. Functional analysis revealed bacterial specialization depending on the type of polysaccharides ingested throughout the weaning transition. Taken together, those results underlined key factors to modulate the gut microbiota trajectories (either maturation rate or endpoints).

## RESULTS

**Voluntary precocious solid food ingestion is modulated by the polysaccharide content of the diet.** We first examined suckling rabbits' feeding behavior to validate the stimulation of early-life solid food ingestion (Fig. 1A). Gel food was consumed from d7 with a 35% higher ingestion of RFF-enriched gels compared to starch-enriched gels (Fig. 1B and C: 5.8 ± 0.4 g of RFF gels/rabbit, i.e., 1.5 g/rabbit in dry matter versus 4.3 ± 0.3 g of STA gels/rabbit, i.e., 1.1 g/rabbit in dry matter). The number of visits to the gel cups counted within two litters indicated that all the kits were responding to the gel stimuli (Fig. S2 in the supplemental material). At d17, gel consumption accounted for 3% of the raw ingestion and 10% of the total dry matter (major supply

from the milk with limited pellet consumption). Altogether, our results confirmed that providing starter food gels reduces the age of first solid food consumption and that early-life solids ingestion is dependent on the polysaccharide content of the meal.

**The timing of solid food introduction and lifelong dietary polysaccharides ingestion slightly affect the host development and immune response.** Afterwards, we analyzed the consequences of our nutritional interventions on the rabbit's growth and health. Overall mortality rate was low with 1.3% of deaths from d3 to weaning and one dead rabbit found afterwards. Live weights were equal between the groups during the suckling period ($403 \pm 5$ g/rabbit at d21), at weaning ($984 \pm 6$ g/rabbit), and during the exclusive solid-feeding period ($1652 \pm 11$ g/rabbit at d50) (Fig. S3). The cecum of the rabbits from the RFF+ group was 32% heavier compared to the STA+ group at d30 ($P < 0.05$) (Table S2).

Systemic and local immune response were assessed by measuring the levels of plasma IgG and cecal IgA and the concentration of hydroperoxydes in serum (Fig. S4). IgA concentrations varied greatly with age, with highest levels at d18 due to passive immunity (i.e., milk IgA), but did not vary with the treatments. Plasma IgG concentrations were lower in the STA+ group compared to the STA group at d30 ($-1.2$ mg/mL, $P < 0.05$) and compared to the RFF+ rabbits at d58 ($-1.2$ mg/mL, $P < 001$). At d30, serum hydroperoxyde levels tended to be lower in the STA+ group than in the STA– group ($P = 0.082$). These results therefore suggest moderate impact on the animal response of early feeding practice or dietary polysaccharide modulations.

**Characteristics of the rabbit hindgut microbiota.** As expected, a shift in microbial composition of the hindgut was observed over time, with marked effects observed when the solid food ingestion sharply increased together with a gradual decrease in milk intake (between d18 and d32). The microbial community structure and composition were in consonance with the sequential microbiota colonization previously observed in the cecum of healthy rabbits (23, 27). All the phyla and families were affected by age. One substantial pattern observed over time was the gradual and significant increase in *Firmicutes* relative abundance (+50% between d18 and d58, $P < 0.001$) together with a decrease in *Bacteroidetes* abundance (–45% between d18 and d58, $P < 0.001$; Fig. 2). In both gut segments, the dominance of the *Firmicutes* was driven by an early increase of *Lachnospiraceae* and *Ruminococcaceae* between d18 and d25 ($P < 0.001$) followed by a stabilization. The proportion of *Bacteroidaceae* sharply decreased between d18 and d25 ($P < 0.001$ in cecum and appendix) and showed a regular depletion afterwards. The differentiation between the cecum and the appendix ecosystem structure was initiated from d58 based on wUniFrac distances (Fig. S5). We will focus below on the effects of early food introduction (comparison STA–/STA+) and diet composition (comparison STA+/RFF+).

**Precocious solid food ingestion and dietary polysaccharides differently affect the developing cecal microbiota. Effects of precocious ingestion of solid food on the cecal microbiota.** To assess effects of postnatal ingestion of solid food on the cecal microbiota, we compared bacterial communities of the groups STA– and STA+ (early access to solid food). The InvSimpson index, which reflects species richness and evenness, was higher in 30-day-old STA+ rabbits ($P < 0.05$; Fig. 3A). Throughout the experiment, the cecal microbiota structure of STA+ rabbits was found closer to the corresponding 58-day-old state compared to the STA– group ($P < 0.01$; Fig. 3B), emphasizing a more mature gut ecosystem when precocious access to solid food was given. The relative abundances of the two main phyla were affected at d30 with 10% less *Bacteroidota* and 10% more *Firmicutes* in the cecum of STA+ rabbits compared to the STA– group ($P < 0.05$; Fig. 3C). At the family level, we observed lower abundances of *Bacteroidaceae* before weaning in STA+ rabbits (Fig. 3D and Table S3A). Six bacterial operational taxonomic units (OTUs) with relative abundance superior to the quantitative threshold of 0.5% were highly differentially abundant ($|\log_2$ fold change$| > 2$) between STA– and STA+ during the preweaning period (Fig. 3E and Table S3B).

**Effects of dietary polysaccharides on the cecal microbiota.** The comparison between STA+ and RFF+ groups was made to assess the effect of the polysaccharide

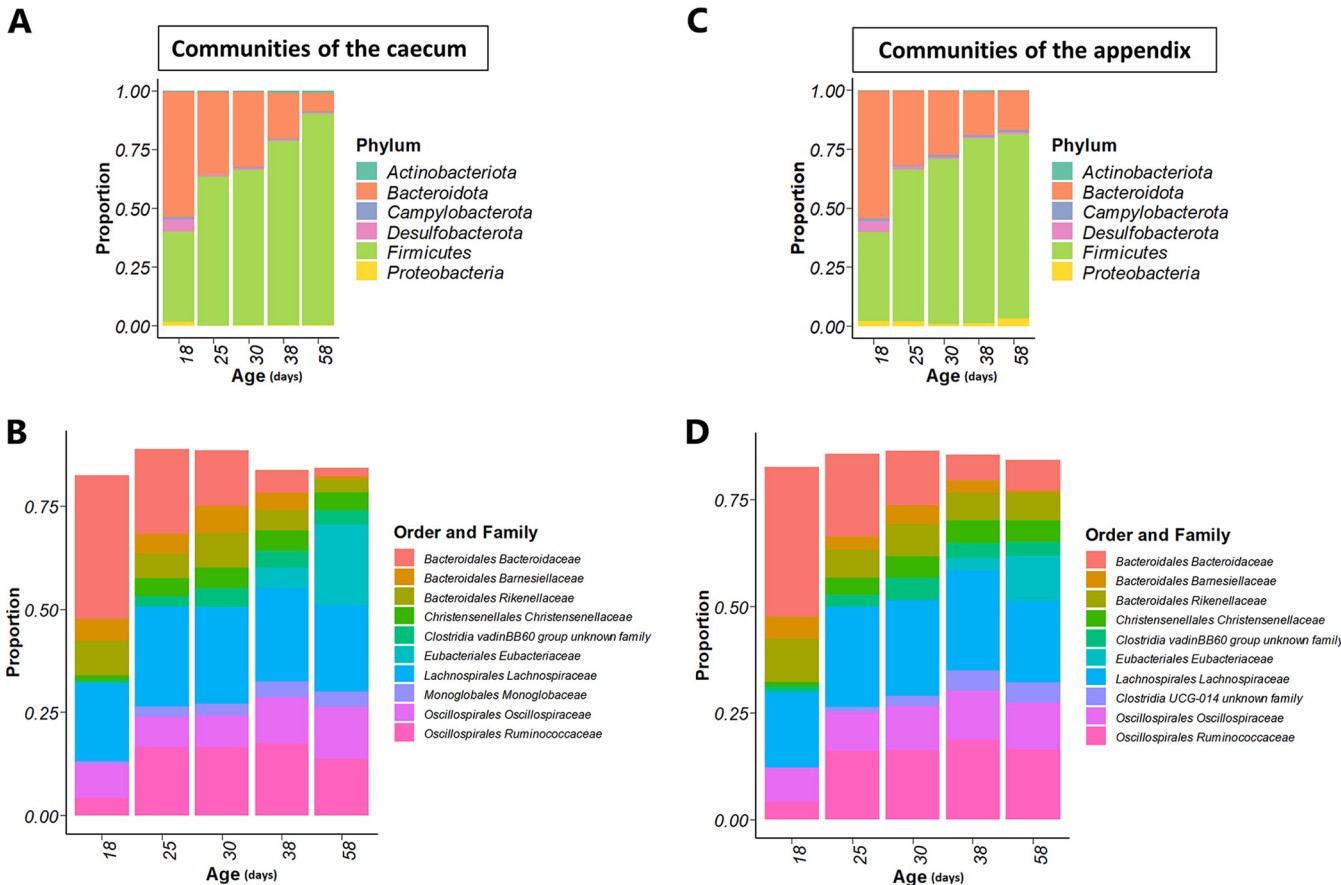

**FIG 2** Distribution of bacterial taxa in the cecum (A, B) and the appendix (C, D) of rabbits regardless of the experimental treatments ($n$ = 27–30 rabbits/age). Relative abundances of phyla (A, C) and the 10 most abundant families (B, D) are presented.

ingestion on the cecal microbiota throughout life. Lower bacterial richness was observed in the RFF+ group rather than in STA+ rabbits at d30, d38, and d58, together with a lower InvSimpson index at d30 and d38 ($P < 0.05$; Fig. 3A). The cecal microbiota structure of RFF+ rabbits was further from the corresponding 58-day-old state compared to the STA+ based on wUniFrac distances calculation ($P < 0.01$ when significant; Fig. 3B). At the phylum level, the ratio *Firmicutes/Bacteroidota*, which is known to increase during the transition from young to adult (28), was lower in the RFF+ group at d38 compared to STA+ (4 versus 7; $P < 0.05$) as a consequence of more abundant *Bacteroidota* (+6% in RFF+ at d38, $P < 0.05$; Fig. 3C). Differences in main family relative abundances were observed, such as lower proportions of *Ruminococcaceae* in the RFF+ group before weaning together with higher relative abundances of *Bacteroidaceae* (Fig. 3D and Table S3C). Twenty abundant bacterial OTUs were highly differentially abundant over time according to the dietary levels of polysaccharides (Fig. 3F). Bacteria from *Ruminococcaceae* and *Lachnospiraceae* families accounted for the majority of the differentially abundant OTUs (Table S3B).

Our results support the idea that precocious solid food introduction and dietary plant polysaccharide content both modulate the establishment of the microbial community in a main site of fermentation. Knowing that the development of lymphoid organs after birth is markedly influenced by microbiota exposure (29), the effects of our treatments on the bacterial colonization of the adjacent appendix were then assessed. To this end, the same methodology presented previously was used.

**The *vermiformis* appendix microbiota is affected by early solid food ingestion and to a lesser extent by the nature of the polysaccharides ingested. Effects of precocious ingestion of solid food on the *vermiformis* appendix microbiota.** The effect of early solid food ingestion on the appendix microbiota alpha diversity was

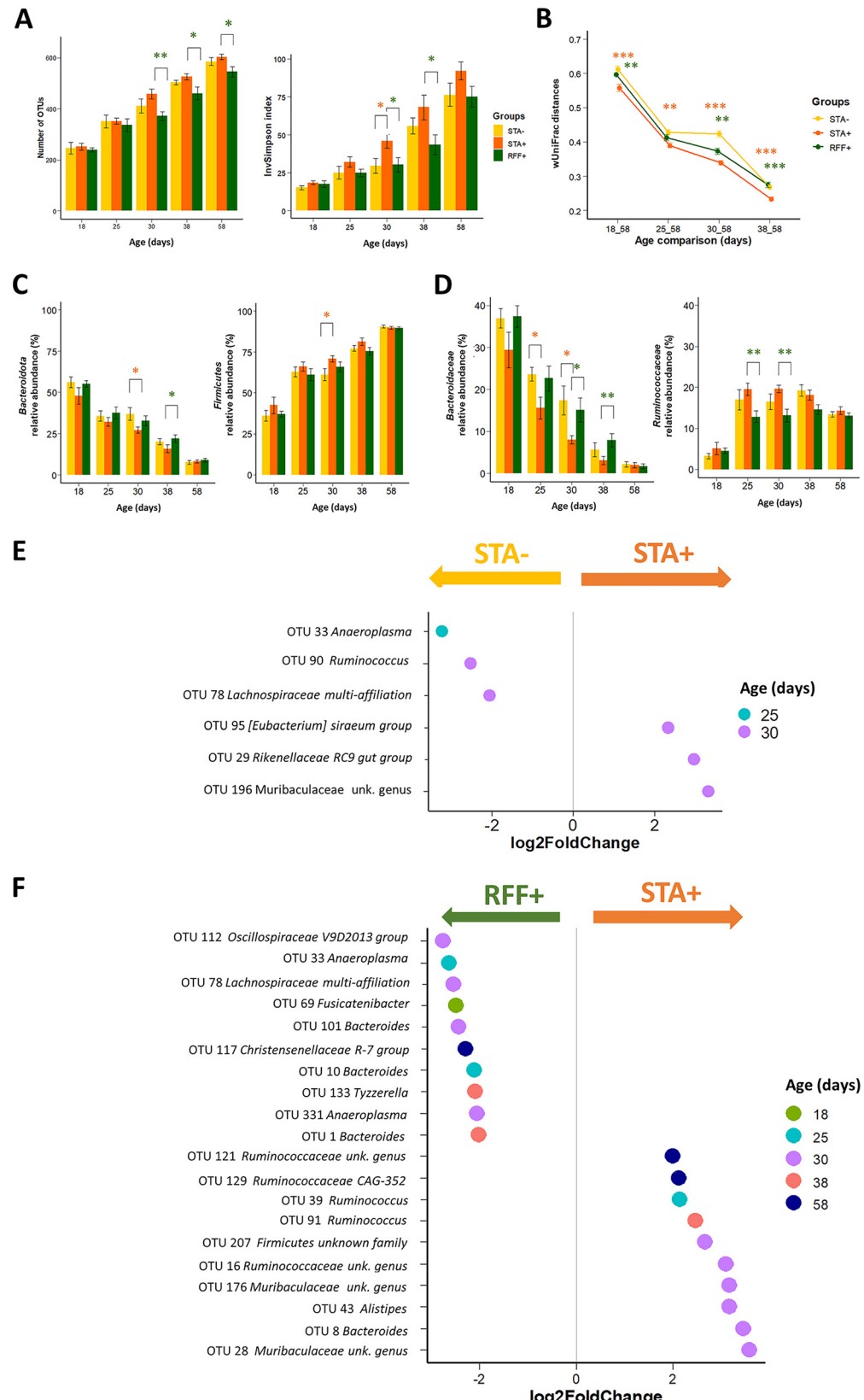

**FIG 3** Analysis of cecal microbiota implantation and colonization according to the experimental groups before and after weaning (Mean ± SEM, n = 9–10 rabbits/age/group). (A) Comparison of alpha–diversity indices. (B) Dissimilarity between the bacterial communities and the 58 days of age state, (C) Distribution of main phyla. (D)

similar to the one observed in the cecum (Fig. 4A). At d18 and d38, the appendix microbiota structure of STA+ rabbits was found closer to the corresponding 58-day-old state compared to the STA– group ($P < 0.01$; Fig. 4B). The changes of the appendix communities at the phyla level (Fig. 4C) resulted in a greater maturity indicator *Firmicutes*/*Bacteroidota* ratio in the STA+ group at d38 compared to STA– treatment (7 versus 4; $P < 0.01$). Around weaning, *Bacteroidaceae* abundances were reduced in the appendix of STA+ rabbits (at d30 STA–: 19% $\pm$ 9%, STA+: 8% $\pm$ 3%; at d38 STA–: 7% $\pm$ 4%, STA+: 3% $\pm$ 1%, $P < 0.01$) (Fig. 4D). Seven OTUs were highly differentially abundant according to the timing of solid food introduction (Fig. 4E).

**Effects of dietary polysaccharides on the appendix microbiota.** Alpha-diversity metrics were not modified by the type of plant polysaccharides consumed, and main taxa abundances only slightly differed between STA+ and RFF+ appendix communities (Fig. 4A, C, and D). Appendix microbiota of 30-day-old and 38-day-old RFF+ rabbits were closer to their corresponding 58-day-old state compared to the STA+ group (Fig. 4B). Ten OTUs from a wide variety of taxa were highly differentially abundant according to diet composition, some of them being affected in the cecum as well (Fig. 4F and Table S3D).

Taken together, those results highlight different responses of the appendix and cecal luminal communities to our two nutritional strategies. Precocious supply of solids affects microbiota composition in both sites, whereas the nature of the dietary polysaccharides has more pronounced effects on the cecal communities.

**Precocious solid food ingestion and dietary polysaccharide type shift the gut metabolomic profile and predicted functionality across time frames.** As a next step, we used NMR-based metabolomics to investigate if the observed modulations of the microbiota composition were associated with adjustments of its metabolic activity. Twenty-nine metabolites were detected in cecal content metabolome from d18. In addition, we inferred the functional profile of cecal bacterial communities. As expected, cecal metabolome was under the influence of age (Fig. S6). We will focus herein on the dietary specific effects.

**Effects of precocious ingestion of solid food on the cecal metabolomic profile and predicted functionality.** The relative concentrations of the luminal cecal acetate and butyrate, as well as the cecal content in amino acids glycine, leucine, lysine, valine, threonine, phenylalanine, and glutamate, increased at d18 when rabbits had precocious access to solid food (Fig. 5A and B; Table S4A). From d25, similar metabolome profiles, pH and $NH_3$ concentrations were observed between the STA– and STA+ groups (Table S2). Based on the metagenomic functional pathways that could be inferred (72% of the ecosystem relative abundances), no differential metabolic potentials were highlighted between STA– and STA+ groups.

**Effects of dietary polysaccharides on the cecal metabolomic profile and predicted functionality.** Ingestion of the RFF-rich diet was associated with decreased relative concentrations of cecal amino acids phenylalanine, valine, as well as glycine at d18 and d58 when compared to the diet enriched in starch (Fig. 5B; Table S4B). Cecal levels of glucose and glutamate were lower in the RFF+ group after weaning. Ingestion of RFF led to the highest cecal relative concentrations of methanol at d18, d25, d30, and d58 (from 1.5- to 2.0-fold increase, $P < 0.001$) and galactose from d25 onwards (from 1.7- to 2.5-fold increase, $P < 0.001$). Cecal acetate content was higher in the cecum of 30-day-old (+35%, $P < 0.05$) rabbits from the RFF+ group compared to the STA+ group (Table S4B). The number of functional pathways inferred from well-covered OTUs was affected by the nature of the dietary polysaccharides from d30.

**FIG 3** Legend (Continued)

*Bacteroidaceae* and *Ruminococcaceae* family abundances. (E and F) $Log_2$ fold change superior to |2| of the OTUs significantly affected by the timing of solid food introduction (STA–/STA+), and dietary polysaccharide ratio (RFF+/STA+) based on differential abundance analysis (DESeq2). "unk. genus" stands for unknown genera. Orange stars represent significant differences induced by precocious solid food introduction (STA–/STA+), whereas green stars emphasize significant differences induced by the food composition (STA+/RFF+) (*, $P < 0.05$; **, $P < 0.01$; ***, $P < 0.001$).

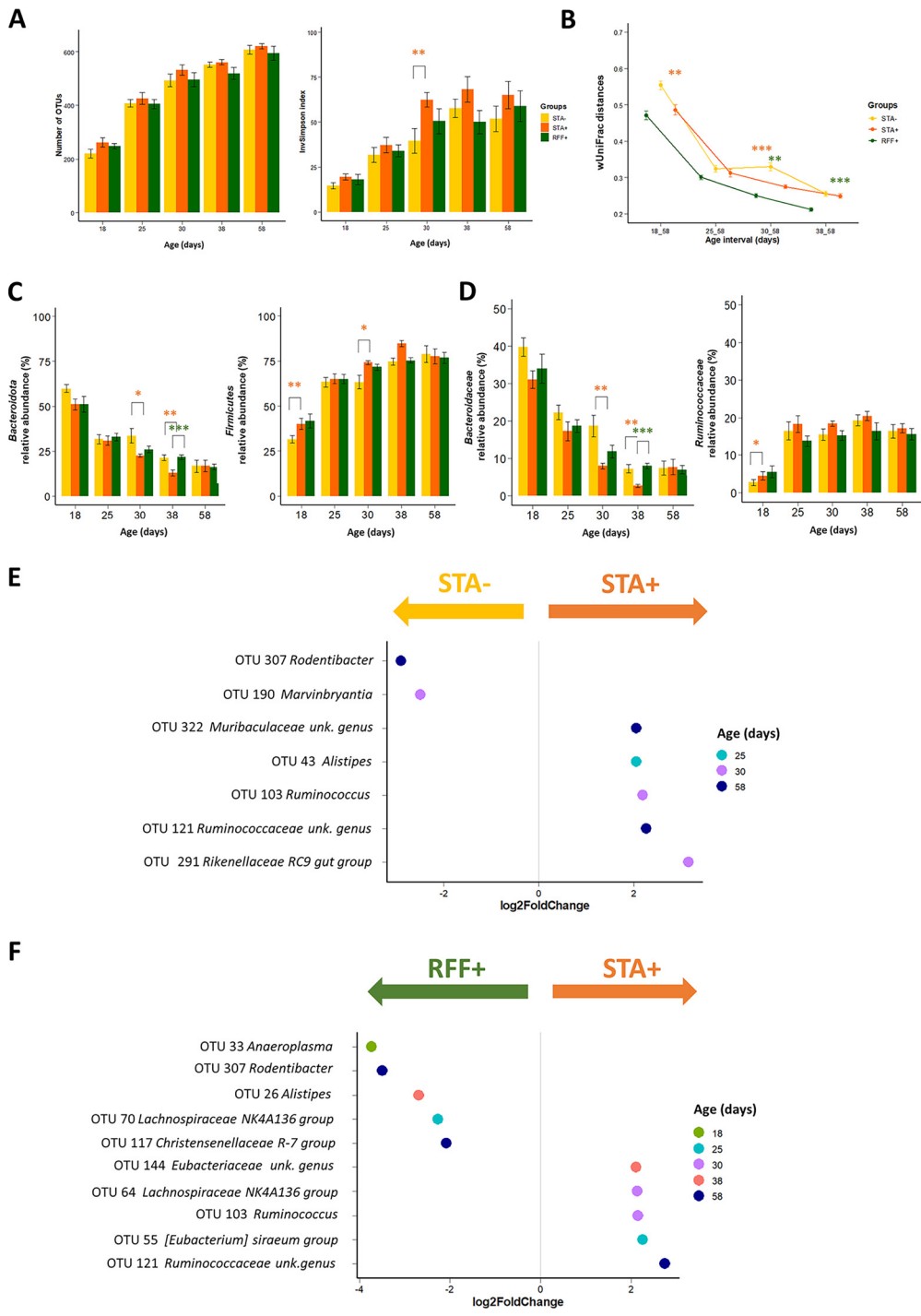

**FIG 4** Analysis of appendix microbiota implantation and colonization according to the experimental groups before and after weaning (Mean ± SEM, $n$ = 9–10 rabbits/age/group). (A) Comparison of alpha-diversity indices. (B) Dissimilarity between the bacterial communities and the 58 days of age state. (C) Distribution of main phyla. (D) *Bacteroidaceae* and *Ruminococcaceae* family abundances. (E and F) Log$_2$ fold change superior to |2| of the OTUs significantly affected by the timing of solid food introduction (STA–/STA+) and dietary polysaccharide ratio (RFF+/STA+) based on differential abundance analysis (DESeq2). "unk. genus" stands for unknown genera. Orange stars represent significant differences induced by precocious solid food introduction (STA–/STA), whereas green stars emphasize significant differences induced by the food composition (STA+/RFF+) (*, $P < 0.05$; **, $P < 0.01$; ***, $P < 0.001$).

Among the 48 metabolic pathways influenced by the type of plant polysaccharides ingested (Table S4C), 3 were highly differentially abundant between STA+ and RFF+ (|log$_2$ fold change| > 1 and $P < 0.01$). The latter were predicted to be associated with hexitol fermentation and aerobic respiration, and they were downregulated with the

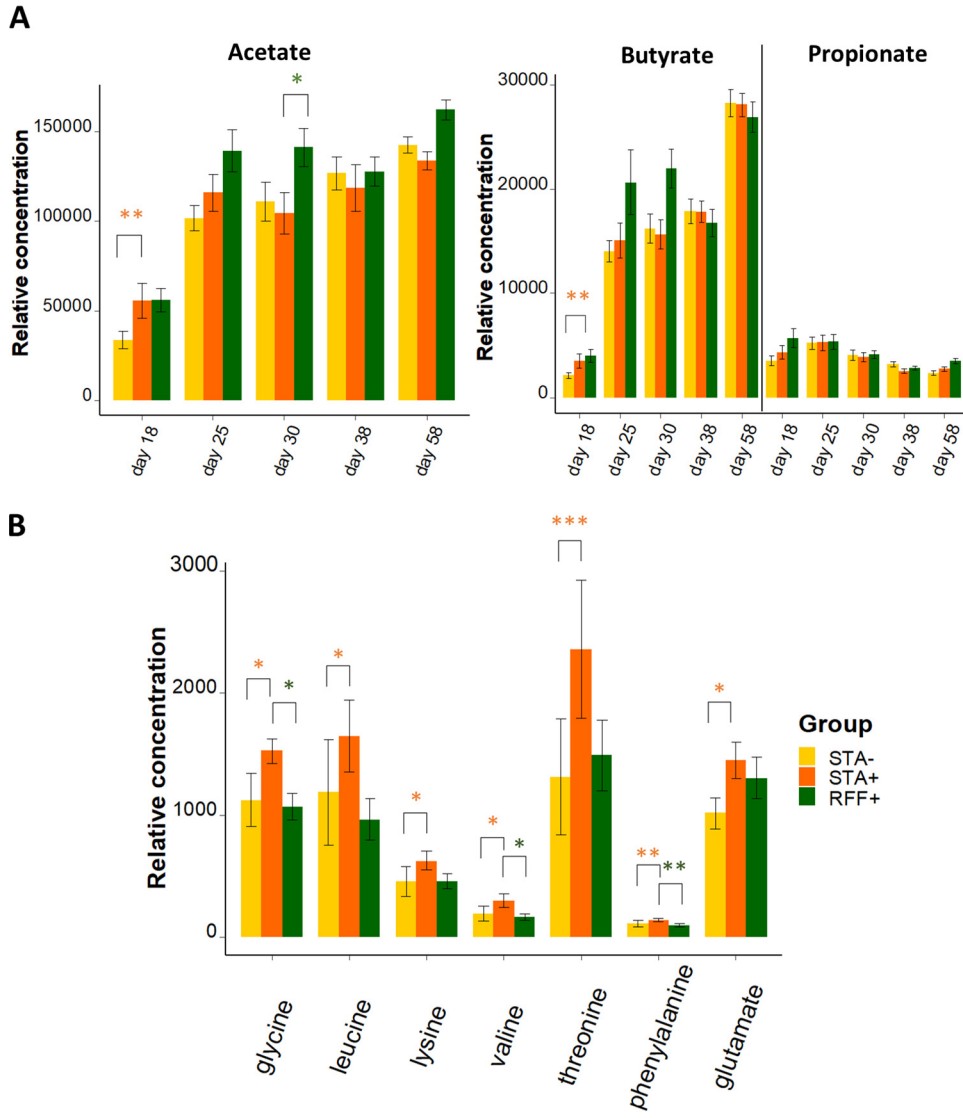

**FIG 5** NMR measurements in cecal luminal content (Mean ± SEM, *n* = 9–10 rabbits/age/group). (A) Relative concentrations of the main cecal short-chain fatty acids throughout the time (B) Relative concentrations of the amino acids affected by one of the nutritional interventions in 18-day-old rabbits. False discovery rate procedure was used to identify differential metabolite levels. Tukey's *post hoc* test was then performed on the selected metabolites. Orange stars represent significant differences induced by precocious solid food introduction (STA−/STA+), whereas green stars emphasize significant differences induced by the food composition (STA+/RFF+) (*, *P* < 0.05; **, *P* < 0.01; ***, *P* < 0.001).

RFF diet. Interestingly, pathways related to plant polysaccharides degradation (L-rhamnose and mannan) were upregulated in the RFF+ group at d38 (*P* < 0.05).

Our results indicate that precocious supply of solid food mainly modified cecal metabolic activities in early life, whereas long-lasting change of diet through different polysaccharide ingestion modulated microbial activities over the time including a reshaping of glycan digestion.

**Precocious solid food ingestion and dietary polysaccharide type had moderate effects on the cecal mucosal transcriptome.** To evaluate the host response to the observed changes in luminal environment, the expression of selected genes in the cecal tissue was assessed. As expected, we highlighted an age-related cecal mucosa maturation with high production of transcripts associated with barrier function in early life (genes encoding for antimicrobials peptides, mucins, and tight junctions) followed by upregulation of genes involved in both innate and adaptative immunity at d58 (redox signaling, cytokines, and IgA transport) (Fig. 6). The expression of the IgA-secretion

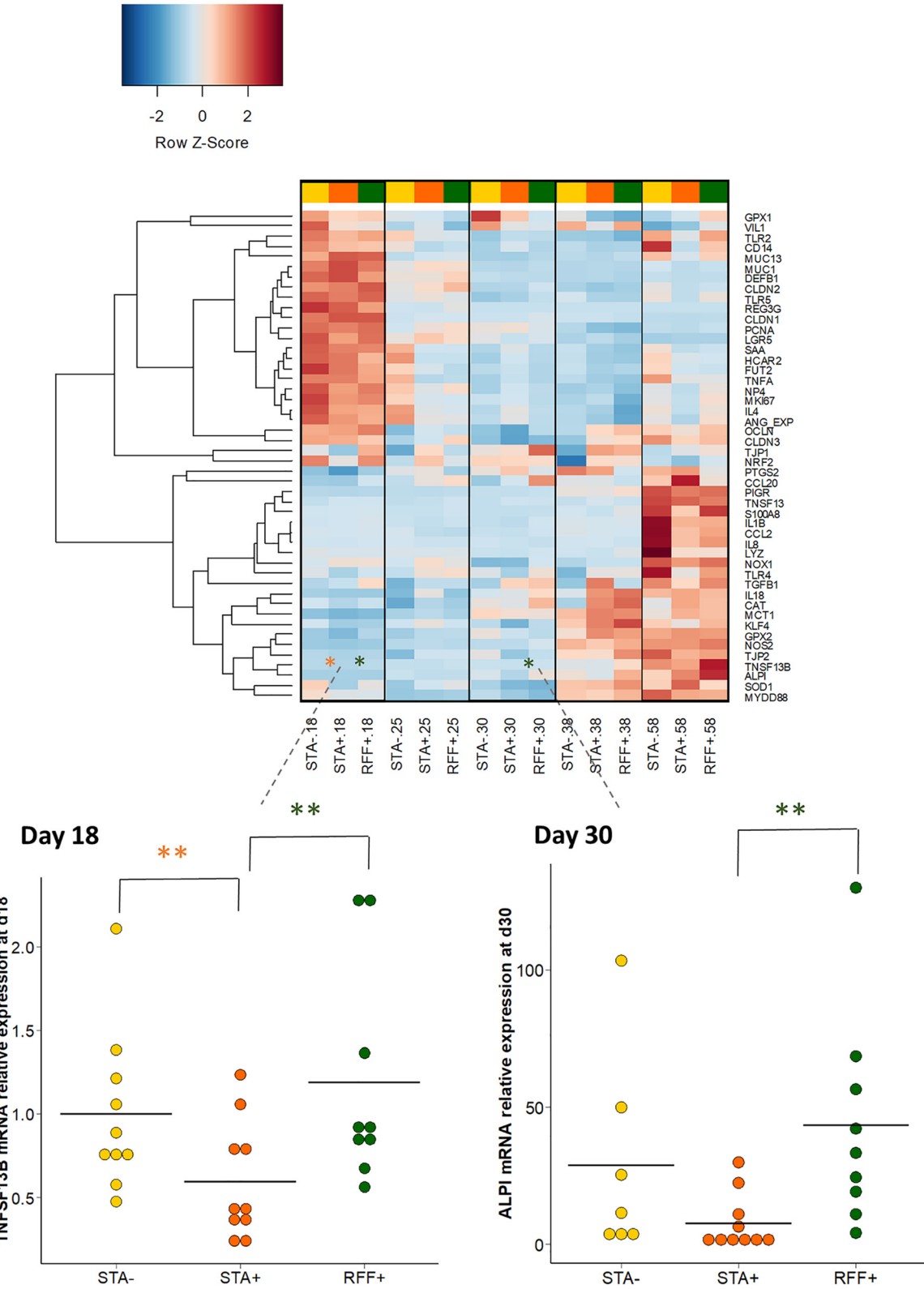

**FIG 6** Screening of the expression of 48 genes of the rabbit cecal mucosa related to health and quantified with microfluidic chip assay. Data were normalized to the average expression in the 18-day-old STA− group. Focus was put on the genes significantly affected by one of the nutritional treatments. False discovery rate procedure was used to identify those genes. Tukey's *post hoc* test was then performed on the selected genes. Orange stars represent significant differences induced by precocious solid food introduction (STA−/STA+), whereas green stars emphasize significant differences induced by the food composition (STA+/RFF+). The horizontal black lines stand for fold change mean values (*, $P < 0.05$; **, $P < 0.01$; ***, $P < 0.001$).

stimulating cytokine *TNFSF13B* (TNF superfamily member 13b) was reduced in STA+ compared to STA– and RFF+ rabbits at d18 ($P < 0.01$, 0.5-fold decrease). At d30, the gene expression of the epithelial differentiation marker *ALPI* (intestinal alkaline phosphatase) was markedly upregulated in cecal mucosa of the RFF+ group compared to STA+ ($P < 0.01$).

## DISCUSSION

Postnatal ingestion of solid food is known to be one major influential factor of microbiota implantation in mammals (11, 30). Recent research suggests that early solids ingestion (i.e., early food diversification) could be beneficial for health (31), presumably through a reshaping of gut microbial colonization (32, 33). Our study aimed to analyze further how early-life ingestion of solid foods affects hindgut microbiota implantation. Two dietary approaches were used to broaden our understanding of microbiota engineering strategies: on one hand, a short and early nutritional intervention was performed to investigate the effect of the timing of solid food introduction, and on the other hand, a long-term dietary modulation enabled us to study dietary polysaccharide effects on the holobiont development.

Overall, precocious ingestion of starch-rich solid food accelerated microbiota maturation in both cecum and appendix. Indeed, when rabbits had early access to solid food enriched in starch, we observed more diverse bacterial communities the week preceding the weaning. Faster stabilization of the microbiota with early feeding practice was also evidenced in the STA+ group compared to the STA–. This is in agreement with previous studies (15, 34, 35, 36) indicating the potential for an earlier introduction of solid feed to reduce the time required for the stabilization of the hindgut bacterial community. Such an enhancement may be advantageous later in life by improving the digestive capacity to cope with dietary shifts (37, 38), by providing the level of microbiota diversity required to inhibit allergic and autoimmune disorders (32), and by stimulating normal components of the immune system (39). But because first gut colonizers play a pivotal role in fermenting the numerous substances present in the milk (40), a precipitated replacement of these bacterial communities may also be detrimental for the host's health, as suggested by prospective human studies (15, 41). When looking at the cecal tissue, decreased expression of the B lymphocytes activator gene *TNFSF13B* was observed at the end of gel supplementation, possibly in response to bacterial signaling (42). However, the downregulation of this gene involved in the adaptive immunity did not modify the cecal IgA content of suckling rabbits. Moreover, it was proposed that a transient reduction of the mucosal IgA content before weaning might be beneficial for the colonization by segmented filamentous bacteria (43, 44). Decreased plasma IgG content after precocious ingestion of starch-rich solid food also pointed out immune modulations due to early solid food introduction. Whether these changes in the humoral immunity can be beneficial for the host needs to be further elucidated. Histological examination of intestinal epithelium, at the interface of the microbiota and the gut mucosal immunity, would help us to investigate further this question. Still, these findings confirmed the importance of early-life solid food ingestion on the host– bacterial symbiosis.

Early ingestion of starch-rich diet promptly stimulated the growth of species belonging to the *Ruminococcaceae* family in the appendix, despite concomitant ingestion of large quantities of milk that might create a niche constraint on the gut microbiome (5, 23, 45). This compositional change is in line with the pattern observed after later introduction of solid food in young mammals (46, 47). The cecal luminal metabolome was also modified after solid food introduction, highlighting a quick adaptation of the developing microbiota activity to new dietary substrates and environmental transitions. Notably, the fermentative activities, assessed by cecal acetate and butyrate concentrations, transiently increased after the ingestion of small quantities of solid food. The levels of several amino acids also increased when plant-based food was introduced in the diet at an early stage. Although those amino acids can be of different origins (48), we hypothesize that poorly digestible plant protein from solid food partly reached the cecum, thus precociously modifying the amino acids content of the hindgut environment, whereas milk proteins are highly

digested in the upper gastrointestinal tract (49). These modulations of the substrates available in the cecal luminal content likely explains some long-term effect of early life solid food ingestion on taxonomic modifications, such as a drop in the *Bacteroidaceae* family, a "milk-oriented microbiota" characteristic taxon (50). After the establishment of the pioneer species, early supply of solid food was thus found to drive the second colonization of gut communities.

Dietary fiber fractions represent major modulators of digestive physiology due to their influence on the nutrient rate of passage, mucosal functionality, and gut microbiota (51). The particularity of this study was to modulate starch and digestible fiber fractions only while maintaining a balanced diet. Our results showed that the type of dietary complex plant carbohydrates shaped both composition and functions of the hindgut microbiota, with major effects observed when the ingestion of solid food became predominant over milk. Not surprisingly, the type of polysaccharides ingested modulated the distribution of species from *Firmicutes* and *Bacteroidota* phyla, which encompass plant-degrading bacteria (52). At the family level, we found that the distinct carbohydrates contents of STA and RFF diets drove a differential establishment of *Lachnospiraceae* and *Ruminococcacae*, presumably because they encompass two plant degrader groups characterized by different carbohydrate-active enzymes within their genomes (53). When rabbits ingested high amounts of starch throughout their lives, the cecal communities reached a mature state faster (27). We hypothesized that resistant starch (54), which has the ability to be quickly fermented and to modulate the gut microbiota (10), may have contributed to this stabilization. Indeed, glucose is readily absorbed in the rabbit small intestine (55), and the higher levels of free glucose observed in the cecum of rabbits that received the enriched-starch diet could be related to the bacterial degradation of starch resistant to host digestion. Diet enrichment with RFF was associated with decreased diversity in the cecum and appendix, with a lower InvSimpson index outlining the dominance of some abundant species in those ecosystems. This can be attributed to the specific dietary content of RFF diet but also can be due to an improved fiber fraction digestibility as reported when increasing digestible fiber to starch ratio (56). Interestingly, higher levels of cecal methanol were found when rabbits received higher quantities of RFF. This must be explained by high dietary concentrations of pectins, a substrate that can be fermented into methanol by specific intestinal bacteria such as members of the *Bacteroides* genus (57). Consistently, the RFF diet was associated with increased cecal acetate content before weaning, in line with the pectin *in vitro* fermentation process assessed in human stools (58). PICRUSt predictions were sketchy and should be treated with caution. However, these predictions seem in line with our hypothesis since increased expression of glycans-degradation pathways was observed in the cecal microbiome of the rabbits that received the RFF diet. In terms of host response, lifelong ingestion of a diet rich in RFF was associated with increased gene expression of *ALPI*, considered as a marker of epithelial differentiation and a regulator of intestinal inflammation (59). Moreover, animals that ingested more RFF had heavier cecum, which may highlight faster development of this organ in response to fermentable nonstarch polysaccharides.

Interestingly, we observed different distribution drivers of microbial populations in the two contiguous digestive segments investigated, even though these sections do not differentiate from each other during the neonatal phase (60). Cecal communities were mostly affected by the type of polysaccharides substrates provided, with pronounced effects on alpha diversity and OTU composition once solid food ingestion overtook milk consumption. In contrast, the composition of the appendix microbiota was substantially influenced by food ingestion stimulation in early life. These differential effects can be linked to the dualism of functions between the cecum and appendix, respectively related to nutrition and immunity (26). Differences in peristaltic movements may also explain these distinct microbiota (61): the narrow lumen of the appendix limits the nutrient flow,

which could induce an inertia of the microbiota. The appendix has long been considered as a degenerating organ, but it could represent a reservoir for beneficial bacteria that can reinoculate the bowel (62). Moreover, the microbial communities of the gut-associated lymphoid tissue are essential for the immune system development (63). We thus believe that the effects of early-life stimuli on the colonization of lymphoid organs deserve increased attention.

We demonstrated that very early ingestion of solid food in infant rabbits, although in small quantities, induced changes in gut microbiota colonization and activity, with an acceleration of the ecological species succession and increased production of short-chain fatty acids. Postnatal solid ingestion showed more impact on the appendix microbiota pattern, whereas the type of dietary plant polysaccharides mainly modified microbiota composition and functions in the cecum. In sum, we showed that the gut microbiota development trajectory was partly modified by the type of plant polysaccharides ingested, whereas its maturation speed was more dependent on the timing of solid food introduction. However, no strong impact on the endpoint studied was achieved, suggesting a dilution of the effects observed at maturity. Those results evidence the possibility of shaping the developing microbiota with nutrition leverage. An important matter to resolve in future research will be to understand the implication of early modifications of the microbiota on the young mammal immune response in a challenging environment.

## MATERIALS AND METHODS

**Animals and experimental design.** Animals were raised and handled at the INRAE experimental unit (PECTOUL, Castanet-Tolosan, France) according to the European Union's recommendations concerning the protection of animals used for scientific purposes (2010/63/EU) and in agreement with the French legislation (NOR: AGRG1238753A 2013). The experiment received the approval of the local ethics committee (SSA_2019_001).

Forty-eight crossbred litters (Hyplus PS19 x Hyplus PS59) were raised until weaning with their doe using a mother–litter separate feeding system (64). Two days after parturition (d2), litter size was standardized to 10 kits. At d3, litters were allocated to three experimental groups (*n* = 16 litters/group), differing by feeding practices After 36 days, kits were weighed, weaned, and moved to cages of 5 rabbits according to their group and litter of origin. A follow-up of individual rabbit weight was then performed at d50.

Controlled suckling was applied once a day in the morning between 0800 h and 0900 h from d2 to d21. Afterwards, free access for nursing was given to the doe rabbits. Milk ingestion was determined as the difference between doe weight before and after nursing at d3, d7, d10, d14, d17, and d21 (65). Litter weight after suckling was recorded at d3, d14, d21, and d28.

Plastic cups were set up within the litter's nest from d3 to d18. They were used as feeders dedicated to the kits and were filled with food in a hydrated gel in two experimental groups. Pelleted food was offered *ad libitum* to all the litters from d15 to weaning, in metal feeders. During the exclusive solid-feeding period (d36–d58), a restricted feeding at 75% of *ad libitum* ingestion was applied, following common breeding practices. Rabbits received water *ad libitum* through water nipples. No antibiotic treatment was administered during the experiment.

**Experimental diets.** Throughout the experiment, rabbits had access to a diet either rich in starch (STA diet) or enriched with rapidly fermentable fibers (RFF diet). In the early feeding groups (recorded as STA+ and RFF+), the litters were fed from d3 to d17 nutritional substrates in a hydrated gel form (to optimize the ingestion [66]) in addition to maternal milk, whereas no gels were distributed to the group STA− (Fig. 7). Gel food ingestion was measured daily from d7 on a dry matter basis. In addition, webcams were used to record within two litters the number of visits to the gels for each kit, as an attempt to assess individuals' feeding behavior. Pelleted food was offered to all the litters from d15 to the end of the experiment. Animals were maintained on the same diet as per their experimental groups (STA or RFF pellets).

Two diets were formulated that differed only in the ratio of RFF to starch (0.6 for the STA diet and 2.0 for the RFF diet; Table S1). The RFF component includes pectins, $\beta$-glucans, fructans and gums. It excludes the Neutral Detergent Fibers (NDF) fraction, i.e., lignin, hemicellulose, and cellulose. From an analytical point of view, RFF fraction represents the Total Dietary Fibers minus the NDF fraction (51). To achieve balanced diet formulation, the same ingredients were used between the two experimental diets with higher levels of cereals (barley, wheat) and sunflower meal in the STA diet and higher concentrations of wheat bran fraction, beet, and grape pulp in the RFF diet.

The food gels were processed with the corresponding kits' pellets (either STA or RFF pellets) transformed into mash supplemented with attractive vanilla aroma (66) (0.06%, Phodé, Terssac, France) and mixed with hot water (75%) and agar (0.6%). In accordance with the preferences of rabbits in the suckling period (64), 2.5-mm-diameter pellets were provided *ad libitum* from d15 to weaning and switched to pellets with a diameter of 4 mm from weaning to d58.

**Sample collection.** At d18, d25, d30, d38, and d58, 10 kits per group were weighed and then euthanized by electronarcosis followed by exsanguination. Those dates correspond to key developmental

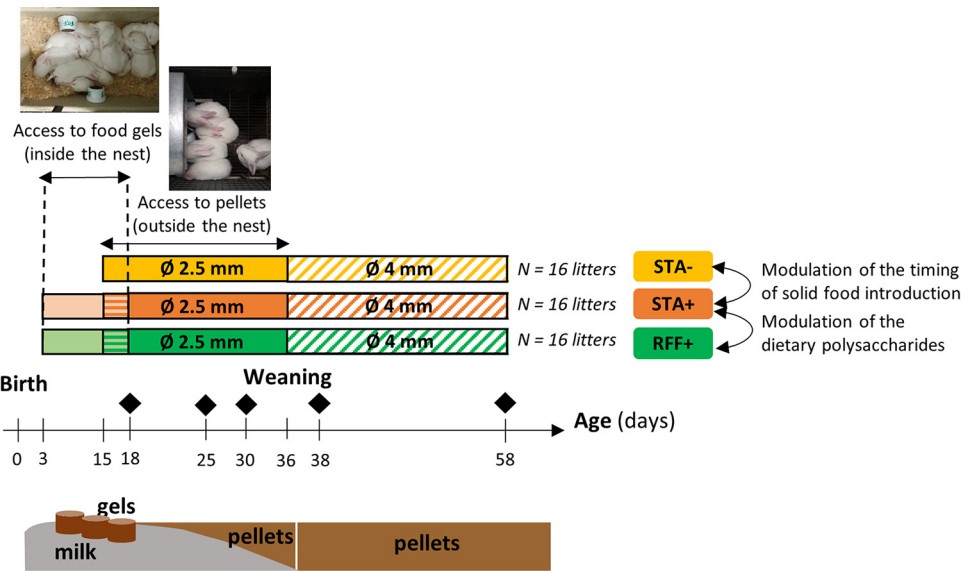

**FIG 7** Experimental design. Before weaning, food was offered *ad libitum*. Afterwards, access to food was restricted (hatched area). Diamonds stand for the sampling dates (*n* = 30 rabbits/date). STA-, access to starch-enriched food in a pellet form from d15 to d58; STA+, access to starch-enriched food from d3 to d18 in a gel form and from d15 to d58 in a pellet form; RFF+, access to food rich in rapidly fermentable fibres from d3 to d18 in a gel form and from d15 to d58 in a pellet form.

stages of the rabbit's digestive bacterial ecosystem (23). At d18, rabbits that visually exhibited an interest in gel food were selected for euthanasia, whereas a random selection was performed at the subsequent dates. Blood samples were collected during slaughter in EDTA tubes and dry tubes for plasma and serum preparation, respectively. Immediately after euthanasia, the cecum, the *appendix vermiformis*, the stomach, and the spleen were isolated and weighed full. Afterwards, cecal and appendix contents were collected in sterile tubes and stored at −80°C until further analysis. Proximal cecum tissue was flash frozen and preserved at −80°C. pH was measured at the junction of the ileum and cecum quickly after slaughter (VWR Collection SP225, Radnor, PA, USA), and additional cecal luminal content was diluted in $H_2SO_4$ (at 2% wt/vol) to quantify ammonia ($NH_3$) concentrations.

**ELISA measurements and evaluation of oxidative stress.** Total plasma IgG and cecal IgA levels were determined in duplicates by sandwich ELISA in 96-well plates coated with specific polyclonal goat anti-rabbit IgG or IgA antibodies (Bethyl Laboratories, Montgomery, TX, USA) with plate reading at 450 nm as previously described (45). IgG was quantified by using a reference IgG serum (Bethyl Laboratories). For IgA analysis, 12 samples were pooled to build a reference for the standard curve construction. Oxidative stress status of the rabbits was assessed on serums by quantification of the derivatives of reactive oxygen metabolites (hydroperoxydes primarily) following the procedure of the d-ROM test (Diacron International, Grosseto, Italy).

**Evaluation of the cecal metabolome.** Ammonia was measured quantitatively with a colorimetric method using a continuous flow analyzer (SAN++, Skalar, Norcross, GA, USA) as previously described (67). Results were expressed according to the liquid phase of the cecal content after determination of its dry matter at 103°C for 24h. Cecal content metabolome was analyzed by using ¹H nuclear magnetic resonance (NMR)-based metabolomics, following the experimental procedure and the data processing previously described (46).

**DNA extraction and 16S rRNA gene sequencing.** Total microbial genomic DNA was extracted from 50 mg of cecal or appendix luminal content using the Quick-DNA Fecal/Soil Microbe 96 Kit (ZymoResearch, Irvine, CA, USA) after mechanical lyses by bead beating, according to manufacturer's instructions. 16S rRNA gene V3-V4 region was amplified using the primer set 343 F/784 R (343 F: 5′–CTTTCCCTACACGACGCTCTTCCGATCTACGGRAGGCAGCAG-3′ and 784 R: 5′–GGAGTTCAGACGTGTGCTCTTCCGATCTTACCAGGGTATCTAATCCT-3′) (68, 69). Thirty PCR amplification cycles were carried out with an annealing temperature of 65°C. The PCR products were controlled by gel electrophoresis. Single multiplexing was then performed using 6bp indexes, this indexes were added to R784 during a second PCR with 12 cycles using forward primer (5′-AATGATACGGCGACCACCGAGATCTACACTCTTTCCCTACACGAC-3′) and reverse primer (5′-CAAGCAGAAGACGGCATACGAGATGTGACTGGAGTTCAGACGTGT-3′). The resulting PCR products were purified and sequenced by MiSeq Illumina Sequencing at the Genomic and Transcriptomic Platform (INRA, Toulouse, France) as previously described (23). PhiX was used as a quality and calibration control for the sequencing run. Negative (ultra-pure water) and positive control (four standard bacterial strains) were sequenced within samples to assess sequencing performance.

**Sequence analysis.** We obtained 12,298,433 16S ribosomal DNA sequences from 296 samples (41,548 ± 11,734 reads per sample). Sequence processing was performed using the Galaxy-supported pipeline FROGS (70). After discarding sequences with primer mismatch and after deleting sequences with erroneous sequencing length (<400 or >500 nucleotides) and ambiguous base,

10,141,290 sequences were left (82% kept, 34,261 $\pm$ 10,458 reads per sample). One low-sequence sample was discarded so that the number of reads per sample ranged between 8,802 and 56,641. Chimeric DNA sequences were detected using VSEARCH (71) and were removed. Clustering was performed using SWARM 21 (aggregation distance of 3) (72). After that, we assigned the operational taxonomic units (OTUs) to the obtained paired sequences using the BLAST algorithm against the SILVA SSU Ref NR 138 data set 22 (73). A total of 1,197 OTUs were kept (min: 224 – max: 927 OTUs per sample). An OTU table with taxonomic affiliations as well as metadata was then built for subsequent analysis with the phyloseq package (74).

**Prediction of microbial functional profiles.** PICRUSt2 analysis was used to predict the cecal microbial community functions with the unrarefied OTU abundance table as input (75). Nearest sequenced taxon index (NSTI) values are an indicator of PICRUSt2 prediction accuracy, with lower scores indicating availability of closely related genomes (75). NSTI scores across all bacterial samples were relatively high (0.42 $\pm$ 0.35). Therefore, OTUs with insufficient reference genome coverage were removed from the analysis, i.e., predictions with NSTI scores superior to 0.5. This threshold was chosen as a compromise between accuracy (removal of samples with low NSTI scores) and reliability (72% of the total relative abundances kept within our microbial communities), as presented in Fig. S1. Eventually, 232 OTU out of the 1,197 identified were used for the prediction. The MetaCyc database was used to classify and interpret the relevant metabolic pathways.

**Cecal tissue gene expressions.** Cecal tissue total RNA extraction was performed in TRI reagent using the Direct-zol kit (ZymoResearch) following the manufacturer's instructions ($n$ = 149 samples extracted). cDNA was prepared from 1 $\mu$g RNA with Superscript II (ThermoFisher Scientific, Waltham, MA, USA) following the manufacturer's instructions. RNA and cDNA quantity and quality were determined using NanoDrop-2000 (Thermo Fisher Scientific). High throughput real-time qPCR was performed using the Biomark microfluidic system with three 96.96 Dynamic Array IFC for gene expression (Fluidigm, San Francisco, CA, USA). The sequences of the primers used are presented in a supplemental table (see table_primer.txt available at https://doi.org/10.15454/QSTXWF). The relative fold changes of target genes expression were calculated with the $2^{-\Delta\Delta Ct}$ method with *GAPDH* as the housekeeping gene and STA– group at d18 as the reference samples.

**Calculations and statistical analysis.** All statistical analyses were performed using R software (version 4.0), with the corresponding code available at https://github.com/paescharlotte/early_life_nutrition_rabbit/. All the analyses were done with linear mixed models including age, experimental group, and their interaction as fixed effects, litter as random effect, and a correction for age heteroscedasticity when necessary (to satisfy the homogeneity of variances assumption). False discovery rate adjustments (BH procedure) were used for multiple testing.

Microbiome, metabolomics, predicted pathways, and host transcriptome data were analyzed with separate investigations of the effect of the timing of solid food introduction (comparisons STA–/STA+ groups) and dietary polysaccharides (comparisons STA+/RFF+ groups) to support eased interpretation. Within-community diversity was evaluated with the R phyloseq package after rarefaction of the OTU table at 8,802 sequences (74). To examine differences in community structure, weighted UniFrac distances (wUniFrac) were calculated on the rarefied count matrix. A PERMANOVA was used to perform pairwise comparison between groups using the "adonis" function of the R phyloseq package. To evaluate community evolution dynamics to reach stability, i.e., maturity, the wUniFrac distances to reach 58-day-old microbiota structures were evaluated at the different sampling points. Bacterial relative abundances at the phylum and family levels were fourth root transformed, whereas metabolites relative concentrations and gene expression data were $\log_{10}$ transformed. Based on the threshold of 0.5% of relative abundances for OTUs quantitative repeatability (14), OTU differential abundance testing was performed using the DESeq2 package with multiple comparison corrections (76). Likewise, the number of MetaCyc metabolic pathways per treatment was analyzed through DeSeq procedure.

**Data Availability.** Amplicon sequences are available online (NCBI accession PRJNA615661). Analysis scripts are available via https://github.com/paescharlotte/early_life_nutrition_rabbit/. Other data used can be accessed at https://doi.org/10.15454/QSTXWF.

## SUPPLEMENTAL MATERIAL

Supplemental material is available online only.

**FIG S1**, DOCX file, 0.1 MB.
**FIG S2**, DOCX file, 0.04 MB.
**FIG S3**, DOCX file, 0.03 MB.
**FIG S4**, DOCX file, 0.1 MB.
**FIG S5**, DOCX file, 0.1 MB.
**FIG S6**, DOCX file, 0.1 MB.
**TABLE S1**, XLSX file, 0.01 MB.
**TABLE S2**, XLSX file, 0.01 MB.
**TABLE S3**, XLSX file, 0.02 MB.
**TABLE S4**, XLSX file, 0.02 MB.

## ACKNOWLEDGMENTS

We gratefully acknowledge M. Moulis and J.-M. Bonnemere for their assistance at the rabbit experimental unit, as well as M. Segura, who contributed to sample preparation, and C. Bannelier, who performed the food analysis. We thank the people of the research teams NED (https://genphyse.toulouse.inra.fr/groups/ned) and SYSED (https://genphyse .toulouse.inra.fr/groups/sysed) for their assistance during data collection. We especially thank C. Knudsen for her careful proofreading. We acknowledge L. Gress and S. Fourre, who performed the quantitative Fluidigm assay. The authors are grateful to the Genotoul bioinformatics platform Toulouse Midi-Pyrenees and the Sigenae group for providing computing and storage resources thanks to Galaxy instance http://sigenae-workbench .toulouse.inra.fr. We thank the metabolomics platform Metatoul-AXIOM in Toulouse.

We declare that we have no conflicts of interest.

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
