## [Reviewer comments · mSystems]

Early introduction of plant polysaccharides drives the establishment of rabbit gut bacterial ecosystems and the acquisition of microbial functions

Charlotte PAES, Thierry GIDENNE, Karine BEBIN, DUPERRAY Joël, Charly GOHIER, Emeline Guené-Grand, Gwénaél REBOURS, Céline Barilly, Béatrice GABINAUD, Laurent Cauquil, Adrien Castinel, Géraldine Pascal, Vincent Darbot, Patrick AYMARD, Anne-Marie DEBRUSSE, Martin Beaumont, and Sylvie COMBES

Corresponding Author(s): Sylvie COMBES, INRAE

Review Timeline:

Submission Date:	March 11, 2022
Editorial Decision:	March 29, 2022
Revision Received:	April 6, 2022
Accepted:	April 19, 2022

Editor: Paul Cotter

Reviewer(s): The reviewers have opted to remain anonymous.

Transaction Report:

DOI: <https://doi.org/10.1128/msystems.00243-22>

March 29, 2022

Dr. Sylvie COMBES
INRAE
UMR 1388
TOULOUSE 31076
France

Re: mSystems00243-22 (Early introduction of plant polysaccharides drives the establishment of rabbit gut bacterial ecosystems and the acquisition of microbial functions)

Dear Dr. Sylvie COMBES:

Thank you for submitting your manuscript to mSystems. We have completed our review and I am pleased to inform you that, in principle, we expect to accept it for publication in mSystems. However, acceptance will not be final until you have adequately addressed the reviewer comments.

Preparing Revision Guidelines

Sincerely,

Paul Cotter

Editor, mSystems

Journals Department
Reviewer comments:

Reviewer #1 (Comments for the Author):

The authors considered all reviewer comments carefully which results in a successful improvement of the manuscript. The study still included a functional prediction analyses and the authors applied the suggestion of the reviewer to use the NSTI score that's good. But this results in even less outcome. Only 1/4 of the OTUs could now be used for the prediction and the lack of information is seen by not even showing the output, Suppl Fig 1 doesn't count for me in fact. I suggest to remove the prediction from the manuscript as it doesn't give any add value to the story.
Line 307 Bacteroidetes change to Bacteroidota

Reviewer #3 (Comments for the Author):

I reviewed an earlier version of this paper and the authors have significantly improved the work since the previous version. I have several minor comments, which should be addressed before the article is ready for publication.

References: All bioinformatic tools and datasets used in this study should be referenced e.g. VSEARCH, SWARM, SILVA.

All figures: Please include the unit of measurement for "Age" in the axis label and legends for all figures as you have in Figure 2 i.e., replace "Age" with "Age (days)".

Ln 297: "significant increase" Is this statistically significant? I cannot see any statistical values in the text or the figure. Please check if these proportion changes are significant and include the information in both the text and Figure 3.

Ln 297-298: The authors present both phylum level and family level analysis of the caecum and appendix communities in Figure 3, but the family level analysis is not even mentioned in the text. In fact, this 4-panel main figure is described in one sentence in the text. And there does appear to be some interesting findings from my initial observation of the data. For example, it appears that the increase gradual increase in Firmicutes was mostly driven by an early increase in Ruminococcaceae and then a later increase in Eubacteriaceae - although this prediction would need to be backed up with data analysis.

Figure 3. Please include panel labels (A, B, C and D), as you have done for the other figures.

PiCrust analysis: The authors have done a good job in addressing my PiCrust comment and mitigating the limitations of using PiCrust on less extensively sampled microbiomes. The compromise that they have chosen between accuracy and data loss seems sensible.

Ln 457-461: Please provide this data in a Supp table i.e., 48 pathways, fold change and appropriate statistics (in this case I think you need to account for multiple testing). Also please provide the statistical significance of these differences in the main text.

Manuscript mSystems00243-22

Response to the editor

Dear Dr. Cotter,

Thank you for giving us the opportunity to submit a revised draft of the manuscript “Early introduction of plant polysaccharides drives the establishment of rabbit gut bacterial ecosystems and the acquisition of microbial functions” for publication in the mSystems journal. We appreciate the time and effort that you and the reviewers dedicated to improve our manuscript.

We have incorporated most of the suggestions made by the reviewers. Those changes are highlighted within the manuscript. Please see below, in blue, for a point-by-point response to the reviewers’ comments and concerns. All page numbers refer to the revised manuscript file with tracked changes in yellow.

Reviewer comments:

Reviewer #1 (Comments for the Author):

The authors considered all reviewer comments carefully which results in a successful improvement of the manuscript.

Thank you very much for your thoughtful comments and efforts towards improving our manuscript.

The study still included a functional prediction analyses and the authors applied the suggestion of the reviewer to use the NSTI score that's good. But this results in even less outcome. Only 1/4 of the OTUs could now be used for the prediction and the lack of information is seen by not even showing the output, Suppl Fig 1 doesn't count for me in fact. I suggest to remove the prediction from the manuscript as it doesn't give any add value to the story.

Thank you for noticing this improvement. We agree with the reviewer that the reduced coverage prediction is unfortunate and should be considered with caution. This limitation was thus mentioned line 482 and better specified lines 378-379.

Although limited, we would like to save these results because (i) they contribute to provide a global assessment of the bacterial ecosystems (335 pathways evaluated from 72% of the total relative abundances), (ii) the accuracy of the remained predictions was proven and allows us a reasonable discussion.

The output of the predictions was shortly described lines 378-380. It was more developed lines 390-395 as recommended. It was also discussed lines 482-484. We believe it provided some insights into bacterial functions together with metabolomics methodology.

A supplemental table with all the pathways affected by the type of dietary polysaccharides was added to provide more outcome (Supplemental Table S4C). No specific figure was presented within the manuscript because it appears non-relevant to us to emphasize further these results.

We further believe that the supplemental figure S1 proposed can provide interesting references for people wishing to follow-up complex bacterial communities, even though this was not in the main scope of our study. We thus would like to maintain it.

Line 307 Bacteroidetes change to Bacteroidota

We do not understand this point since it appears to us that the correct taxonomic name was used. To be clearer, a mention to the taxonomic level considered was added line 318.

Reviewer #3 (Comments for the Author):

I reviewed an earlier version of this paper and the authors have significantly improved the work since the previous version.

I have several minor comments, which should be addressed before the article is ready for publication.

We thank the referee for the careful and insightful review of our manuscript.

References: All bioinformatic tools and datasets used in this study should be referenced e.g. VSEARCH, SWARM, SILVA.

The corresponding references were added accordingly, i.e, references [36], [37] and [38].

All figures: Please include the unit of measurement for "Age" in the axis label and legends for all figures as you have in Figure 2 i.e., replace "Age" with "Age (days)".

As recommended, the unit was added to the figures 1, 3, 4, 5, 6 and 7 accordingly. Supplemental figures S4,S5 and S6 were also modified to add "(days)" mention.

Ln 297: "significant increase" Is this statistically significant? I cannot see any statistical values in the text or the figure. Please check if these proportion changes are significant and include the information in both the text and Figure 3.

Indeed, our analysis were mainly performed within groups with no emphasize on the age effect to focus on the main scope of the study. To support these conclusions, extra statistical tests were done and are presented lines 298-301. This result was not added within figure 3 because it is not adapted to evidence the numerous age-related changes of bacterial distribution.

Ln 297-298: The authors present both phylum level and family level analysis of the caecum and appendix communities in Figure 3, but the family level analysis is not even mentioned in the text. In fact, this 4-panel main figure is described in one sentence in the text. And there does appear to be some interesting findings from my initial observation of the data. For example, it appears that the increase gradual increase in Firmicutes was mostly driven by an early increase in Ruminococcaceae and then a later increase in Eubacteriaceae - although this prediction would need to be backed up with data analysis.

We agree with the reviewer that this figure could be more extensively described. However, to limit the size of our result sections, we prefer to limit this investigation. We only added the analysis of the longitudinal trends regarding *Lachnospiraceae*, *Ruminococcaceae* and *Bacteroidaceae* to shed light on the next result sections since these families are mentioned. These modifications appear from lines 302 to 304. For more detailed analysis, readers can access to the references quoted [23, 42].

Figure 3. Please include panel labels (A, B, C and D), as you have done for the other figures.

This was added in the figure 3 and corresponding caption.

PiCrust analysis: The authors have done a good job in addressing my PiCrust comment and mitigating the limitations of using PiCrust on less extensively sampled microbiomes. The compromise that they have chosen between accuracy and data loss seems sensible.

Thank you for your feedback on this aspect.

Ln 457-461: Please provide this data in a Supp table i.e., 48 pathways, fold change and appropriate statistics (in this case I think you need to account for multiple testing). Also please provide the statistical significance of these differences in the main text.

We added this detailed information in the Supplemental Table S4C. This allowed us to capture interesting modulations lines 394-395 and these points were more discussed lines 482 to 484. Significant levels were mentioned in the manuscript lines 392 and 395.

April 19, 2022

Dr. Sylvie COMBES
INRAE
UMR 1388
TOULOUSE 31076
France

Re: mSystems00243-22R1 (Early introduction of plant polysaccharides drives the establishment of rabbit gut bacterial ecosystems and the acquisition of microbial functions)

Dear Dr. Sylvie COMBES:

Your manuscript has been accepted, and I am forwarding it to the ASM Journals Department for publication. For your reference, ASM Journals' address is given below. Before it can be scheduled for publication, your manuscript will be checked by the mSystems production staff to make sure that all elements meet the technical requirements for publication. They will contact you if anything needs to be revised before copyediting and production can begin. Otherwise, you will be notified when your proofs are ready to be viewed.

Publication Fees:

We recognize that the video files can become quite large, and so to avoid quality loss ASM suggests sending the video file via <https://www.wetransfer.com/>. When you have a final version of the video and the still ready to share, please send it to mSystems staff at mSystems@asmusa.org.

For mSystems research articles, if you would like to submit an image for consideration as the Featured Image for an issue, please contact mSystems staff at mSystems@asmusa.org.

Sincerely,

Paul Cotter
Editor, mSystems

Journals Department
Fig. S5: Accept
Fig. S1: Accept
Fig. S6: Accept
Table S3: Accept
Table S4: Accept
Fig. S2: Accept
Fig.S4: Accept
Table S2: Accept
Fig. S3: Accept
Table S1: Accept